

# A continuum model for meltwater flow through compacting snow

Colin R. Meyer[1] and Ian J. Hewitt[2]

[1]John A. Paulson School of Engineering and Applied Sciences, Harvard University, Cambridge, MA 02138 USA
[2]Mathematical Institute, Woodstock Road, Oxford, OX2 6GG

*Correspondence to:* Colin R. Meyer (colinrmeyer@gmail.com)

**Abstract.** Meltwater is produced on the surface of glaciers and ice sheets when the seasonal surface energy forcing warms the snow to its melting temperature. This meltwater can run off the surface in streams or percolate through the porous snow and refreeze, which warms the subsurface through the release of latent heat. We model the percolation process from first principles using a continuum model that includes heat conduction, meltwater percolation and refreezing, as well as mechanical compaction. The model is forced by surface mass and energy balances. When the surface temperature reaches the melting point, we compute the amount of meltwater produced and allow it to percolate through the snow according to Darcy's law, or to run off the surface if the snow is already saturated. The model outputs the temperature, density, and water content profiles as well as the surface runoff and water storage. We compare the propagation of freezing fronts that occur in the model to observations from the Greenland ice sheet. The model applies to both accumulation and ablation areas and allows for a transition between the two as the surface energy forcing varies. The largest firn temperatures occur at intermediate values of the surface forcing when perennial water storage is predicted.

## 1 Introduction

The percolation zone of glaciers and ice sheets may play an important role in buffering changes to climate by storing meltwater in the firn and modulating the glacier surface temperature. Meltwater stored in the firn does not run off the surface of the ice sheet and contribute to sea level rise and is not immediately routed to the bed, thereby delaying the influence of meltwater on ice dynamics. Therefore, the capacity of firn to store water is an important metric in assessing the health of glaciers and ice sheets under atmospheric warming (Enderlin et al., 2014; Forster et al., 2014; Machguth et al., 2016). The percolation of meltwater through firn also changes the thermal structure at the surface of a glacier or ice sheet, including the mean firn temperature, which is required as a boundary condition for ice sheet models.

As the snow at the surface is melted, the liquid water holds a substantial quantity of latent heat. If the water runs off through supraglacial streams or drains to the bed through moulins then the latent heat is carried away, leaving relatively cold ice. Thus, instead of warming the ice through the addition of sensible heat, the surface energy flux is converted into latent heat, which drains away as runoff. On the other hand, if the meltwater percolates into the snow and refreezes, it releases the latent heat and warms the snow. Humphrey et al. (2012) observe that the snow at 10 m depth in Greenland is often greater than 10°C warmer than the mean annual air temperature because of the refreezing of meltwater. Thus, the contribution of surface runoff to sea level rise and the near-surface temperature structure is tied to the fate of the meltwater.



In this paper, we focus on the dynamics of meltwater percolating through porous snow. Our approach is to construct a continuum model along the lines of Gray (1996), as opposed to the cell-based numerical models that have so far been applied to the Greenland ice sheet. A commonly used model is the IMAU-FDM described by Ligtenberg et al. (2011) and Kuipers Munneke et al. (2014, 2015). That model includes mechanical compaction and an empirical 'tipping bucket' hydrology scheme, where the firn is divided into distinct layers and water fills each layer up to the irreducible water content and then trickles instantaneously into the lower layers. Runoff occurs when the water reaches an impermeable layer and the water is removed (representing the lateral flow that occurs in reality). Steger et al. (2017a, b) use a similar tipping-bucket method in the SNOWPACK model and compare the results to IMAU-FDM. Additionally, there are several more theoretical models for the percolation of meltwater through porous snow. Colbeck (1972, 1974, 1976) studies the problem of a propagating front of refreezing meltwater, analytically deriving the rate of refreezing and the speed of the front; we will make use of a similar solution as a test of our numerical method. Gray and Morland (1994, 1995) as well as Gray (1996) clarify the previous analyses in the context of mixture theory.

In section 2, we construct our continuum model for the firn layer. We include equations for temperature, porosity, saturation, and ice and water velocities. The meltwater percolation is described using Darcy's law and we distinguish between partially and fully saturated pore space. In the partially saturated regions, the flow is driven primarily by gravity and, to a lesser extent, by capillary pressure gradients. Meltwater percolation is influenced by refreezing which changes the porosity as well as temperature. We unify the temperature, compaction, and meltwater percolation physics into an enthalpy method, which we solve numerically using a finite volume code implemented in MATLAB. The enthalpy method applies to both cold and temperate regions, the boundaries of which it automatically locates (Aschwanden et al., 2012).

One of the difficult aspects of modeling firn in a percolation zone is that both mechanical compaction and refreezing combine to control the changes in snow density. There are various empirical parametrizations of dry compaction that can be used, but their appropriateness for snow containing meltwater is uncertain. The parametrizations represent the rearrangement and growth of snow crystals and the accompanying closure of air voids as functions of temperature and accumulation rate, and these processes may be modified by the presence of liquid between crystals. In the absence of a more developed theory for wet compaction, we take the approach of using these dry parametrizations, but modify the material density derivative to include the rate of refreezing that is calculated from the thermodynamics.

In section 3, we analyze two test problems. First, we look at a front of water propagating through cold snow. The refreezing of the meltwater warms the snow and the front propagates at a constant velocity, which we determine analytically. The speed of the front depends on the water flow rate, the snow porosity, and the initial snow temperature. The results compare favorably with the temperature data from Humphrey et al. (2012). Second, we examine the dynamics when the snow becomes fully saturated with water, i.e. where all of the void space between snow crystals is taken up by liquid water and the air is squeezed out. In this case, there are saturation fronts and again we are able to calculate approximate analytical solutions for their location.

In section 4, we impose a more realistic surface energy forcing to examine the effect of climate variables on the firn hydrology. We look for periodic solutions under seasonally varying forcing. For cold temperatures, this results in the classical 'thermal wave' (Cuffey and Paterson, 2010) but for larger forcing, the temperature profiles are strongly dependent on the meltwater propagation and are intricately tied to the compaction, which controls the porosity. With a large enough surface energy





input, sufficient meltwater is produced for water to exist year-round in the snow, i.e. a perennial firn aquifer (Forster et al., 2014). The year-round presence of liquid water keeps the bottom of the firn warm as compared to the surface of the ice sheet, where the temperature drops below zero in the winter. Our model allows us to determine the parameters under which this type of behavior occurs, and to examine how the mean firn temperature gets colder again as the surface forcing further increases

and a percolation zone transitions to an ablation area. The water that cannot drain into the snow leaves the surface as runoff, which we also compute.

## 2 Model

### 2.1 Percolation through porous ice

Here we describe our model for the flow of meltwater through porous, compacting snow. We keep track of the flow of water,

the compaction, and the melt/refreezing of water into the snow. A volume fraction $1-\phi$ is solid ice while the void space $\phi$ is composed of water and air. We define the saturation $S$ as the fraction of the void space that is filled by water (see schematic in figure 1). Conservation of mass for ice, water, and air are expressed as

$$\frac{\partial}{\partial t}\left[(1-\phi)\rho_i\right] + \boldsymbol{\nabla}\cdot\left[(1-\phi)\rho_i\boldsymbol{u}_i\right] = -m, \tag{1}$$

$$\frac{\partial(S\phi\rho_w)}{\partial t} + \boldsymbol{\nabla}\cdot(S\phi\rho_w\boldsymbol{u}_w) = m, \tag{2}$$

$$\frac{\partial}{\partial t}\left[(1-S)\phi\rho_a\right] + \boldsymbol{\nabla}\cdot\left[(1-S)\phi\rho_a\boldsymbol{u}_a\right] = 0, \tag{3}$$

where the subscripts $i$, $w$, and $a$ indicate ice, water, and air, respectively. The densities $\rho_i$, $\rho_w$, and $\rho_a$ are constants. The variable density of the snow is $(1-\phi)\rho_i + \phi S\rho_w + \phi(1-S)\rho_a$. The rate at which ice melts and turns into meltwater internally is given by $m$, and is therefore a source in equation (2) and a sink in equation (1). This term is always negative, i.e. refreezing, and in fact is zero, except on refreezing interfaces. We assume that the air density is negligible and henceforth neglect equation (3).

The flow of water is governed by Darcy's law, i.e.

$$\phi S\left(\boldsymbol{u}_w - \boldsymbol{u}_i\right) = -\frac{k(\phi)}{\mu}k_r(S)\left(\boldsymbol{\nabla}p_w + \rho_w g\hat{\boldsymbol{z}}\right), \tag{4}$$

where $p_w$ is the water pressure, $k(\phi)$ is the permeability, $k_r(S)$ is the relatively permeability, and $\mu$ is the viscosity of the water. For the permeability we use a simplified Carman-Kozeny relationship, given by

$$k(\phi) = \frac{d_p^2}{180}\phi^3 = k_0\phi^3, \tag{5}$$

where $d_p$ is a typical grain size. See table 1 for the parameter values we use later.

We must now distinguish between partially saturated ($S < 1$) and fully saturated ($S = 1$) flow. When the snow is partially saturated, capillary forces drive flow along liquid bridges connecting ice crystals (Bear, 1972). Thus, we relate the water pressure to the capillary pressure $p_c$ by $p_w = p_a - p_c$, where $p_a$ is the air pressure (taken as zero). Both the capillary pressure

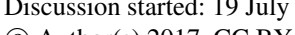



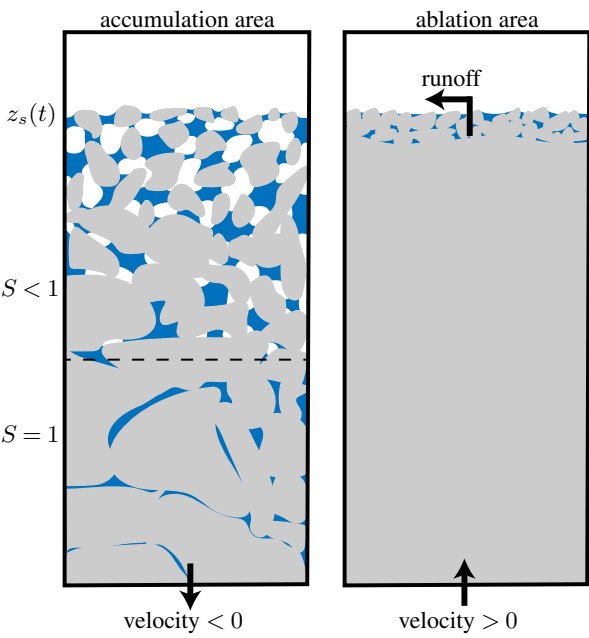

**Figure 1.** The three components of meltwater-infiltrated snow: air, water, and ice. The left panel shows water infiltrating an accumulation area where the snow density increases with depth and snow advects down. The water partially saturates the snow near the surface ($S < 1$) whereas, at depth, all of the air is squeezed out and the snow is fully saturated ($S = 1$). The right panel shows an ablation area where the there is fully saturated porous snow in a thin layer near the surface and the underlying ice is solid, advecting into the domain from upstream. Ice grains make contact in the third dimension and similarly many of the air and water pockets are connected in the third dimension.

and the relative permeability are prescribed functions of the saturation $S$. We take

$$k_r(S) = S^\beta \quad \text{and} \quad p_c(S) = \frac{\gamma}{d_p} S^{-\alpha}, \tag{6}$$

where $\gamma$ is surface tension, and choose the exponents $\alpha$ and $\beta$ such that $\beta = \alpha + 1$, which avoids a singularity in $k_r(S)p_c'(S)$ at $S = 0$ (Gray, 1996).

5    If the snow is fully saturated, water pressure $p_w$ is constrained by mass conservation. Combining equations (1), (2), and (4) gives

$$\nabla \cdot \left[ \boldsymbol{u}_i - \frac{k(\phi)}{\mu} \left( \rho g \hat{\boldsymbol{z}} + \nabla p_w \right) \right] = m \left( \frac{1}{\rho_w} - \frac{1}{\rho_i} \right), \tag{7}$$

which is an elliptic problem for $p_w$. Boundary conditions for this equation are provided by the constraint that $p_w$ must be continuous across the interfaces between partially and fully saturated regions and the constraint of no flow across impermeable 10    boundaries (e.g. ice lenses).





## 2.2 Compaction

For dry snow, typical compaction models relate the rate of change of density, or equivalently porosity, to quantities such as depth, accumulation rate, temperature, and grain size. In our context, these can be expressed using the material derivative

$$\frac{\partial \phi}{\partial t} + \boldsymbol{u}_i \cdot \boldsymbol{\nabla} \phi = -\mathscr{C}, \tag{8}$$

where $\mathscr{C}$ is a parametrization of the rate of compaction (Arthern et al., 2010). We will assume that these compaction processes are unaltered by the presence of meltwater but must augment equation (8) with the additional contribution to changes in porosity due to refreezing. Therefore, we take

$$\frac{\partial \phi}{\partial t} + \boldsymbol{u}_i \cdot \boldsymbol{\nabla} \phi = \frac{m}{\rho_i} - c\phi, \tag{9}$$

where the specific compaction rate we choose is the Herron and Langway (1980) model, which is written as $\mathscr{C} = c\phi$. The coefficient $c$ in units of $\text{yr}^{-1}$ is given by

$$c = \begin{cases} 11a\exp\left\{-\frac{1222}{T}\right\} & \text{if } \phi > 0.4 \\ 575\sqrt{a}\exp\left\{-\frac{2574}{T}\right\} & \text{if } \phi \le 0.4 \end{cases}, \tag{10}$$

where $a$ is the accumulation rate in meters of water equivalent per year and $T$ is the absolute temperature. This is an empirical parametrization where the two forms reflect a change in dominant compaction processes at a certain snow density. Other parametrizations for compaction that could easily be incorporated in this framework are discussed by Zwally and Li (2002), Reeh (2008), and Morris and Wingham (2014).

Combining with (1), we note that (9) is equivalent to

$$(1-\phi)\frac{\partial w_i}{\partial z} = -c\phi. \tag{11}$$

## 2.3 Temperature

We assume that ice and water are at the same temperature and therefore any region containing meltwater ($S > 0$) is at the melting point $T_m$. In regions without water, we solve the temperature evolution equation,

$$\rho_i c_p (1-\phi)\frac{\partial T}{\partial t} + \rho_i c_p (1-\phi)\boldsymbol{u}_i \cdot \boldsymbol{\nabla} T = \boldsymbol{\nabla} \cdot \left(\overline{K}\boldsymbol{\nabla} T\right) - \mathscr{L}m, \tag{12}$$

where $\overline{K} = (1-\phi)K$. The latent heat term $-\mathscr{L}m$ operates on interfaces of refreezing, where it is singular and causes discontinuities to occur in the temperate gradient.

## 2.4 Surface boundary conditions

Here we write boundary conditions on the surface $z_s(t)$, which we assume is locally flat, and we write $w_i$ and $w_w$ as the vertical velocities. The kinematic conditions are

$$\rho_i(1-\phi)(w_i - \dot{z}_s) = \rho_w(M - a), \tag{13}$$

$$\rho_w \phi S(w_s - \dot{z}_s) = \rho_w(M - R + r), \tag{14}$$



where $\dot{z}_s$ is the velocity of the surface, and $M$ is the rate of melting, $a$ is the accumulation rate, $R$ is the rainfall rate, and $r$ is runoff, all expressed in units of water equivalent per year. The compaction equation (9) requires a boundary condition, $\phi = \phi_0$, when the accumulation rate is greater than the rate of melting (*i.e.* $w_i - \dot{z}_s < 0$), where $(1-\phi_0)\rho_i$ is the density of freshly deposited snow. The energy balance on the surface provides a boundary condition for the temperature equation when

the surface temperature is less than $T_m$, and determines the rate of melting $M$ when $T = T_m$. These conditions are combined as

$$\rho_i c_p \left(1 - \phi\right)\left(w_i - \dot{z}_s\right)\left(T - T_m\right) - \overline{K}\frac{\partial T}{\partial z} = -Q + h(T - T_m) + \rho_w \mathscr{L} M, \tag{15}$$

along with the conditions $M = 0$ when $T < T_m$, and $M \geq 0$ when $T = T_m$. The forcing flux $Q(t)$ includes the combined effects of radiative, turbulent, and sensible heat fluxes. We assume that this is prescribed in order to provide a simple parametrization

of the climate forcing. However, it can be related to more specific components of the energy balance, see Appendix A. The heat transfer coefficient $h$ represents a combination of radiative and turbulent heat transfer. We expect $Q$ to have a typical magnitude on the order of $Q_0 = 200$ W m$^{-2}$ with a comparable seasonal amplitude, and take $h = 14.8$ W m$^{-2}$ K$^{-1}$ as a representative constant (Cuffey and Paterson, 2010; van den Broeke et al., 2011).

## 2.5 Numerical method

Our complete model is given by ice and water mass conservation (1) and (2), Darcy's law (4), compaction (9), and temperature evolution (12), subject to the boundary conditions (13)-(15). The model is forced by a prescribed energy flux $Q$, accumulation $a$, and precipitation $R$, and it predicts the temperature, porosity, and saturation profiles as well as the surface melt rate, runoff, refreezing, and storage of liquid water.

    In this section, we rewrite the equations in a form that we use for our numerical solutions. There are two steps: first, we

combine the equations as conservation equations for total water (ice and liquid water) and enthalpy (sensible and latent heat). Using this approach, commonly referred to as the enthalpy method, we can avoid tracking the phase change interfaces and can solve for their location using inequalities (Hutter, 1982; Aschwanden et al., 2012; Hewitt and Schoof, 2017). The second step is to change variables into a frame that moves with the ice surface. At this stage we also simplify the model to write it in one vertical dimension, and we make the Boussinesq approximation to ignore density differences so that $\rho_i = \rho_w = \rho$.

We define the total water as $\mathcal{W}$, which is the sum of liquid and solid fractions, i.e.

$$\mathcal{W} = 1 - \phi + S\phi, \tag{16}$$

and we define the enthalpy as the sum of sensible and latent heat as

$$\mathcal{H} = \rho c_p \mathcal{W}(T - T_m) + \rho\mathscr{L}S\phi, \tag{17}$$

The inverse relationships that relate the enthalpy $\mathcal{H}$ and total water $\mathcal{W}$ to the temperature $T$, saturation $S$, and porosity $\phi$ are

$$T = T_m + \min\left\{0, \frac{\mathcal{H}}{\mathcal{W}}\right\}, \quad \phi = 1 - \mathcal{W} + \max\left\{0, \frac{\mathcal{H}}{\rho\mathscr{L}}\right\}, \quad \text{and} \quad S = \max\left\{0, \frac{\mathcal{H}}{\rho\mathscr{L}\phi}\right\}. \tag{18}$$





We define the depth below the ice surface $Z$, and the relative downward ice velocity $\widetilde{w}_i$, as

$$Z = z_s(t) - z, \qquad \widetilde{w}_i = \dot{z}_s - w_i. \tag{19}$$

We combine the conservation equations (1), (2), with Darcy's law (4), and temperature evolution (12) as

$$\frac{\partial \mathcal{W}}{\partial t} + \frac{\partial}{\partial Z}\left(\widetilde{w}_i \mathcal{W} + q\right) = 0, \tag{20}$$

$$\frac{\partial \mathcal{H}}{\partial t} + \frac{\partial}{\partial Z}\left\{\widetilde{w}_i \mathcal{H} + q\left[\rho c_p(T - T_m) + \rho \mathscr{L}\right] - \overline{K}\frac{\partial T}{\partial Z}\right\} = 0, \tag{21}$$

where the downward water flux is

$$q = \frac{k(\phi)k_r(S)}{\mu}\left(\rho g - \frac{\partial p_w}{\partial Z}\right) \tag{22}$$

Combining ice conservation (1) with compaction (9) gives

$$(1 - \phi)\frac{\partial \widetilde{w}_i}{\partial Z} = -c\phi, \tag{23}$$

and water pressure is given by

$$p_w = -\frac{\gamma}{d_p}S^{-\alpha} \quad (S \leq 1) \quad \text{or} \quad p_w \geq -\frac{\gamma}{d_p} \quad (S = 1). \tag{24}$$

The surface boundary conditions ($Z = 0$), re-expressed in terms of $\mathcal{W}$ and $\mathcal{H}$, are given as

$$\mathcal{W}\widetilde{w}_i + q = a + R - r, \tag{25}$$

$$\widetilde{w}_i \mathcal{H} + q\left[\rho c_p(T - T_m) + \rho \mathscr{L}\right] - \overline{K}\frac{\partial T}{\partial Z} = Q - h(T - T_m) + \rho\mathscr{L}R - \rho\mathscr{L}r, \tag{26}$$

$$\widetilde{w}_i = \frac{a - M}{1 - \phi_0} \quad (\widetilde{w}_i > 0) \quad \text{or} \quad \widetilde{w}_i = \frac{a - M}{1 - \phi} \quad (\widetilde{w}_i \leq 0), \tag{27}$$

In these conditions, the runoff $r$ is assumed to be zero unless the snow at the surface reaches full saturation, in which case, equations (25) and (26) determine $r$. At the bottom of our domain, we assume that the conductive heat flux and pressure gradients vanish to replicate effective matching conditions to the deep interior of the ice sheet. On internal interfaces between fully saturated and partially saturated regions we apply $p_w = -\gamma/d_p$ to ensure pressure continuity.

We discretize the conserved fluxes in space using a finite volume method implemented in MATLAB (see Supplementary Information for code). In this construction, the value of each variable is constant in each cell center and the velocities and fluxes are evaluated at cell edges, thereby transferring fluxes of each variable from one cell to another. We evolve equations (20)-(21) in time using explicit forward Euler timestepping, which involves evaluating the fluxes on the cell edges using the quantities from the previous timestep. For advection, we use an upwinding scheme where the value of the variable advected depends on the velocity direction. For edges between partially saturated cells, we evaluate the water fluxes using the capillary pressure for $p_w$ on the adjacent cells. For edges between fully saturated cells, we solve equation (20) with $S = 1$ as an elliptic equation for $p_w$ on the saturated cells, which we then use to evaluate the water fluxes. In order to allow cells to switch from fully to partially saturated, we compute the fluxes using both of these methods on edges between fully and partially saturated cells and choose that which gives the largest flux away from the saturated region.



| $\rho$ | 917 kg m$^{-3}$ | $Q_0$ | 200 W m$^{-2}$ | $\mathcal{U}$ | 100 |
|---|---|---|---|---|---|
| $c_p$ | 2050 m$^2$ s$^{-2}$ K$^{-1}$ | $h$ | 14.8 W m$^{-2}$ K$^{-1}$ | $\mathcal{S}$ | 12 |
| $\mathscr{L}$ | 334,000 m$^2$ s$^{-2}$ | $\Delta T$ | 13.5 K | $\mathcal{B}$ | 260 |
| $K$ | 2.1 kg m s$^{-3}$ K$^{-1}$ | $\mathcal{M}$ | $6 \times 10^{-4}$ kg m$^{-2}$ s$^{-1}$ | $Pe$ | 11 |
| $g$ | 9.806 m s$^{-2}$ | $\ell$ | 20.6 m | $\alpha$ | 1 |
| $\gamma$ | 0.07 N m$^{-1}$ | $k_0$ | 5.6$\times 10^{-11}$ m$^2$ | $\beta$ | 2 |
| $d_p$ | 10$^{-4}$ m | $t_0$ | 3.15$\times 10^7$ s | | |
| $\mu$ | 10$^{-3}$ Pa s | | | | |

**Table 1.** Table of physical values, derived scales, and nondimensional parameter values (defined in Appendix B).

## 3  Test problems

In this section we consider two test problems that demonstrate the model behavior and validate the numerical method. The two problems that we consider here are designed to explore the boundaries between frozen and unfrozen snow (refreezing interfaces) as well as the boundaries between partially and fully saturated snow (saturated interfaces). We start by describing
the propagation of rain water into dry snow. This is similar to the problem studied by Colbeck (1972), Gray (1996), and Durey (2014) and has an approximate analytical solution that provides a useful test case for the enthalpy method. We also compare the results of the analytical solution for the propagation of the meltwater front to temperature data from Humphrey et al. (2012). Secondly, to test the propagation of saturated fronts, we consider an isothermal problem in which the porosity profile is prescribed to decrease with depth. We again investigate how rain water propagates into the snow, with saturation increasing
as the front propagates down. At a certain point the snow fully saturates and a saturated front propagates up toward the snow surface.

### 3.1  Rainfall into cold snow

We consider the infiltration of rain into cold, dry snow as a test problem. We start with a patch of dry snow ($S = 0$) with constant porosity ($\phi = \phi_0$) and temperature ($T = T_\infty < T_m$). We assume no accumulation and ignore compaction so that the
ice is stationary. Furthermore, the porosity is large enough that the snow never fully saturates. At time $t = 0$ a fixed flux of rain $R$ with a temperature $T = T_m$ is applied at the surface $Z = 0$ and a wetting front at $Z = Z_f$ moves down at velocity $\dot{Z}_f$ (we show a schematic in figure 2 and the numerical solutions in figure 3). Since the capillary pressure gradients are small and the flow is largely driven by gravity, the wetting front behaves as a smoothed shock front. Some of the water at the shock front refreezes, warming the snow ahead. As shown in more detail in Appendix C, the behavior of this shock can be understand
by ignoring the diffusive capillary pressure term. This approximation relegates equations (1) and (2) to hyperbolic partial

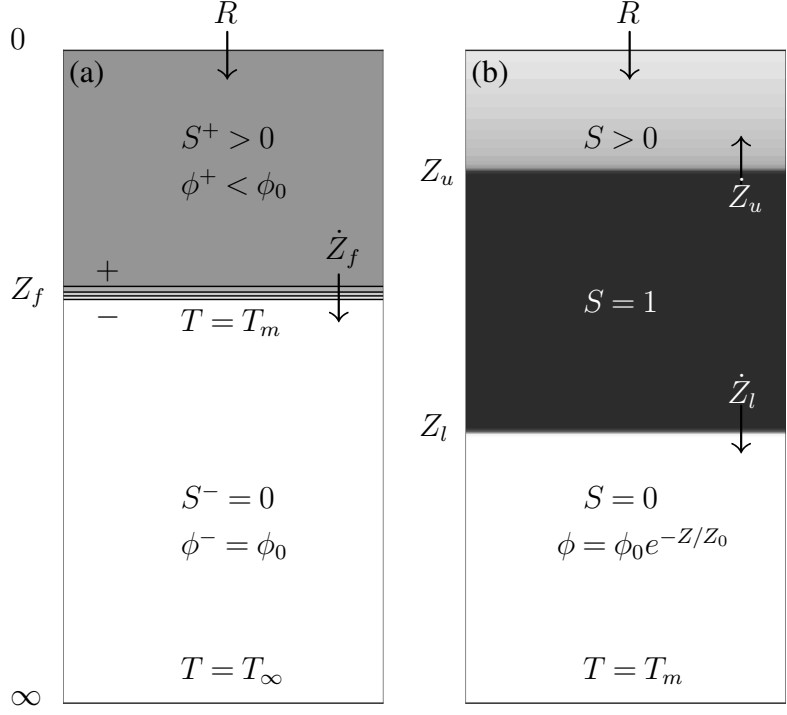

**Figure 2.** Schematic of the test problems considered in (a) section 3.1 and (b) section 3.3. In both panels, rain falls at a rate $R$ on the surface of the snow. In panel (a), the snow is initially cold $T = T_\infty$ and dry $S = 0$ snowpack with porosity $\phi_0$. The rain water percolates through the snow, refreezes at the interface $Z_f(t)$, and releases latent heat that warms the snow. The refreezing decreases the porosity in the upper region so that $\phi^+ < \phi_0$. In panel (b), the snow is temperate $T = T_m$ with a porosity profile that decays exponentially with depth. After the snow fully saturates two saturation fronts emerge with $\dot{Z}_l$ propagating downward and $\dot{Z}_u$ upward.

differential equations for the porosity and saturation as well as simplifying the temperature equation (12) so that

$$\frac{\partial S}{\partial t} + \frac{\rho g k(\phi) k_r'(S)}{\phi \mu} \frac{\partial S}{\partial Z} = 0, \quad (0 < Z < Z_f) \tag{28}$$

$$\frac{\partial \phi}{\partial t} = 0, \quad (0 < Z < Z_f) \quad \text{and} \quad (Z > Z_f) \tag{29}$$

$$\frac{\partial T}{\partial t} = \frac{K}{\rho c_p} \frac{\partial^2 T}{\partial Z^2}, \quad (Z > Z_f) \tag{30}$$

5    with initial and boundary conditions

$$S = 0, \quad \phi = \phi_0, \quad T = T_\infty \qquad \text{at} \qquad t = 0, \tag{31}$$

$$T = T_\infty \qquad \text{as} \qquad Z \to \infty, \tag{32}$$

$$T = T_m \qquad \text{on} \qquad Z = Z_f. \tag{33}$$

$$\frac{\rho g k(\phi) k_r(S)}{\mu} = R \qquad \text{on} \qquad Z = 0, \tag{34}$$




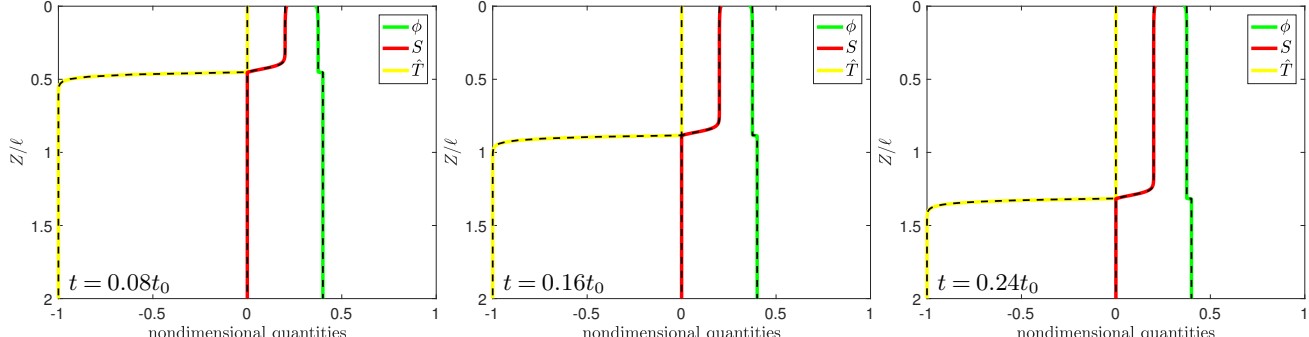

**Figure 3.** Evolution of a refreezing front at three instances of time, partitioned between the three components of the enthalpy. The green, red, and yellow colors show the porosity, saturation, and temperature profiles, respectively. The dashed lines show the approximate analytical solutions described in Appendix C. The temperature is made nondimensional by $T = T_m + (T_\infty - T_m)\hat{T}$ and the parameters are $\phi_0 = 0.4$, and $R = 0.54$, along with other values in table 1.

Equations (28)-(30) have corresponding jump conditions across the shock which incorporate the refreezing rate $-m_I$ at that front. These are

$$\frac{\rho g}{\mu} k(\phi) k_r(S) - \phi S \dot{Z}_f \bigg|_{+} = -m_I, \tag{35}$$

$$\dot{Z}_f [\phi]_{-}^{+} = m_I, \tag{36}$$

$$(1-\phi) K \frac{\partial T}{\partial Z} \bigg|_{-} = \rho \mathscr{L} m_I, \tag{37}$$

where $+$ refers to the region above the front ($Z < Z_f$). Using these jump conditions and the solutions to (28)-(30) subject to the boundary conditions (31)-(34), we find an approximate expression for the front velocity,

$$\dot{Z}_f = \frac{R\mathscr{L}}{\phi^+ S^+ \mathscr{L} + (1-\phi_0)(T_m - T_\infty)}. \tag{38}$$

Note that if $T_\infty = T_m$, i.e. isothermal snow, the front simply propagates at the speed of the draining rain water $R/(\phi^+ S^+)$ (Bear, 1972). The effect of refreezing due to $T_\infty < T_m$ is to slow the front and to cause a decrease in the porosity as the front passes. We also determine approximate analytical solutions for temperature and saturation, which are compared to the numerical solutions in figure 3. The agreement between the numerical and approximate solutions is very good. The approximate temperature profile ahead of the refreezing front is given by

$$T = T_\infty + (T_m - T_\infty) \exp\left[-\frac{\rho c_p \dot{Z}_f (Z - Z_f)}{K}\right]. \tag{39}$$

## 3.2 Data comparison

The refreezing and release of latent heat as a front of meltwater moves through a firn layer allows the percolation of meltwater to be observed in englacial temperature data. Harper et al. (2012) and Humphrey et al. (2012) collected temperature data in the



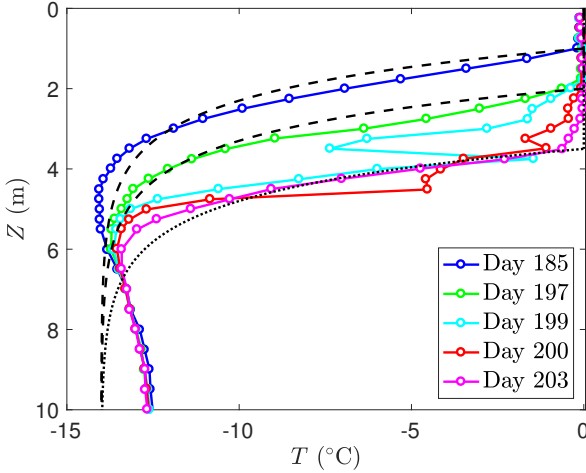

**Figure 4.** Data from Humphrey et al. (2012) show the propagation of refreezing fronts in Greenland firn. We overlay the approximate temperature solution for the temperature ahead of a refreezing front (black lines, equation (39)). The speed of the front varies over the 18-day record: dashed lines use the initial speed and the dotted line uses the final speed. The far-field temperature is assumed to be constant in the model whereas the data show a local minimum in temperature at around 5 m, which could be due to prior freezing fronts or the seasonal wave.

percolation zone on the western flank of the Greenland ice sheet and inferred the movement of meltwater by warming of the snow due to the release of latent heat. They set up a vertical string of thermistors to determine the temperature profile in the upper 10 m of the ice sheet. Data from one vertical string between the dates of 5 July 2007 and 25 July 2007 (days 185-203) is shown in figure 4. From these data it is clear that the ice at depth progressively warmed, likely due to the refreezing of

5   liquid meltwater. Over the twelve days between day 185 and day 197, the warming front propagated about a meter, while over the course of the next six days from day 197 to day 203, the meltwater penetrated two additional meters, showing a four-fold increase in front velocity. Humphrey et al. (2012) infer that the warming spike on day 199 is due to an influx of meltwater from lateral sources. A minimum temperature is observed at around 5 m depth and the temperature recorded on the lower thermistors is warmer, which could be due to prior warming by meltwater pulses or a manifestation of the seasonal thermal wave.

10     We now compare these data to the approximate solution for the temperature field ahead of a refreezing front, as given in equation (39). We fit two different front speeds $\dot{Z}_f$ for the days 185-197 and days 197-203, respectively. We set the melting temperature $T_m = 0$ °C, fit a constant far-field temperature $T_\infty$, and use the heat diffusivity for ice $K/(\rho c_p) = 1.1 \times 10^{-6}$ m$^2$ s$^{-1}$ (table 1). In light of the simplified analysis, the fit between equation (39) and the Humphrey et al. (2012) data is quite good.



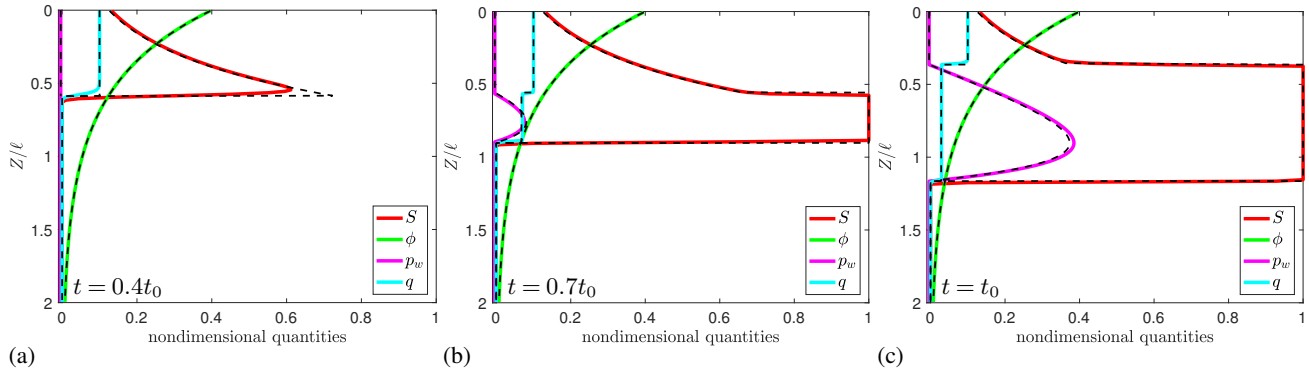

**Figure 5.** Evolution of fully saturated fronts at three instances in time, showing saturation (red), water flux (cyan), and water pressure (magenta). The porosity (green) decreases exponentially with depth over length scale $Z_0 = \ell/2$, where $\ell$ is the characteristic length scale defined in Appendix B and given in table 1. Panel (a) shows the position of the front before the firn fully saturates. Panels (b) and (c) show the bidirectional motion of the fully saturated fronts. Dashed black lines show semi-analytical solutions from solving equation (41).

## 3.3 Isothermal saturation fronts

We now consider the propagation of rain water into isothermal, temperate snow of decreasing porosity such that fully saturated fronts develop. The porosity decreases exponentially with depth as

$$\phi(Z) = \phi_0 e^{-Z/Z_0}, \tag{40}$$

where $Z_0$ is a constant. We continue to ignore compaction and accumulation and since the snow is isothermal, the porosity is therefore constant in time. Initially, the rain partially saturates the snow and a wetting front moves downward, see figure 5(a). Then, at a certain depth, the maximum saturation reaches unity and two saturation fronts emerge, one that propagates up and the other down, see figures 5(b) and 5(c).

In Appendix D, we derive the locations of the upper $Z_u$ and lower $Z_l$ fronts by neglecting flow driven by gradients in capillary pressure. This analysis results in two ODEs for the evolution of upper front $Z_u$ and lower front $Z_l$ as

$$\dot{Z}_l = \frac{q_s}{\phi_l} \quad \text{and} \quad \dot{Z}_u = \frac{q_s - R}{\phi_u \left[1 - \left(\frac{\mu R e^{3Z_u/Z_0}}{\rho g k_0 \phi_0^3}\right)^{1/\beta}\right]} \quad \text{with} \quad q_s = \frac{3 k_0 \phi_0^3 \rho g (Z_u - Z_l)}{\mu Z_0 \left[e^{3Z_u/Z_0} - e^{3Z_l/Z_0}\right]}, \tag{41}$$

subject to the initial conditions

$$Z_u = Z_l = Z_1 \quad \text{at time} \quad t = t_1, \tag{42}$$

which are the location and time at which full saturation initiates. We solve these coupled, nonlinear ODEs using a numerical integrator in MATLAB, and compare these semi-analytical solutions to the full numerical solutions in figure 5 (the dashed black lines). The slight differences are due to neglecting the gradient in capillary pressure in our approximate solutions.



## 4 Results

We now examine the solutions to the full model with seasonal forcing, which we parametrize as a sinusoid, using the annual mean surface energy flux as a control parameter. In principle, we could also incorporate diurnal periodicity, however we choose to ignore it because we expect diurnal variability to affect only a small surface layer and we are interested in the full firn column.

For cold ice, the variation of surface energy flux leads to a seasonal wave temperature structure (Cuffey and Paterson, 2010), and a dry compaction density profile. This solution breaks down when the surface temperature reaches the melting point, at which point the surface snow melts and the meltwater can percolate through the snow and refreeze, thereby warming the snow through the release of latent heat. Even with a small amount of melting, the resulting temperature profiles become very different from the thermal wave.

We apply an oscillating surface forcing in equation (26) of the form

$$Q(t) = \overline{Q} - Q_0 \cos(2\pi t/t_0), \tag{43}$$

where $\overline{Q}$ is the annual mean surface forcing and we take the amplitude $Q_0 = 200 \, \text{W m}^{-2}$, and period $t_0 = 1$ year. For simplicity, we assume a constant accumulation rate and ignore rain fall.

We now run a suite of numerical simulations varying the accumulation rate and annual mean surface forcing, each time allowing the dynamics to reach an annual periodic state (typically this takes around 10 years). Four representative space-time diagrams of these simulations are shown in figure 6. Each case shows a different value of $\overline{Q}$ with the same accumulation rate (1.7 meters water equivalent per year) and porosity of fresh snow $\phi_0 = 0.64$. While the ice surface moves up and down during the simulation, we plot the quantities as a function of depth below the surface $Z = z_s - z$, and plot the ice streamlines to show the relative motion of the ice.

The four simulations in figure 6 represent the spectrum of possible surface types on glaciers and ice sheets, within accumulation, percolation, and ablation regions. If we interpret increasing $\overline{Q}$ as a parametrization of climate warming, we might expect a location that is initially an accumulation area to transition through each of these states. Figure 6I is an accumulation area where there is no melting at any point during the year. The ice streamlines show that the ice advects downward as more snow accumulates on the surface. The snow compaction is visible from a convergence of the streamlines with time. The temperature with depth in this case is just the thermal wave and the variations in surface temperature are only felt around $\sqrt{K/(\rho c_p \omega)} \sim 6$ m into the snow.

Increasing $\overline{Q}$ above -$Q_0$ leads to melting during summer. Figure 6II shows a percolation zone where there is still net accumulation over the year but where the temperature and porosity profiles are significantly affected by the meltwater that drains into the snow during the summer. Here there is water below 10 m throughout the year fed by percolation each summer. This is a perennial aquifer, as found in a number of field observations (Humphrey et al., 2012; Forster et al., 2014; Machguth et al., 2016).

Figure 6III shows a region which is still an accumulation area but with more melting. Interestingly, this situation no longer has a perennial aquifer and all of the meltwater that is produced refreezes. Although still a percolation zone it is different in character than the region shown in Figure 6II. Despite more water being produced on the surface, the larger quantity of water







**Figure 6.** Space-time diagrams showing the evolution of porosity $\phi$ (top), saturation $S$ (middle), and temperature $T$ (bottom) as a function of time for the accumulation rate $a = 1.7$ mwe yr$^{-1}$ and four values of forcing: (I) cold accumulation zone where the mean forcing is $\overline{Q} = -Q_0$. (II) percolation zone with mean forcing $\overline{Q} = -0.707Q_0$. In this case, a clear perennial aquifer develops. (III) a percolation zone with larger forcing $\overline{Q} = -0.575Q_0$. (IV) ablation zone with mean forcing $\overline{Q} = -0.146Q_0$. In all simulations the porosity of the falling snow is $\phi_0 = 0.64$ and the black lines show ice streamlines.





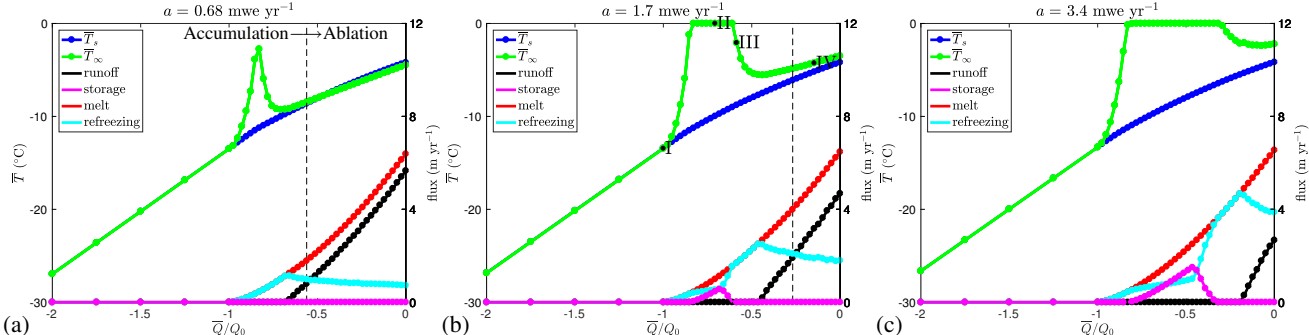

**Figure 7.** Average meltwater partition (right) and annual mean temperature at the ice surface $\overline{T}_s$ and bottom of the domain $\overline{T}_\infty$ (left) as a function of the annual mean surface forcing, with accumulation increasing from left to right: (a) $a = 0.68$ mwe yr$^{-1}$, (b) $a = 1.7$ mwe yr$^{-1}$, and (c) $a = 3.4$ mwe yr$^{-1}$. For $\overline{Q} > -Q_0$ melting occurs at the surface and meltwater percolation warms the bottom of the domain. Dashed lines mark the transition from an accumulation to ablation zone and the roman numerals in (b) correspond to the solutions in figure 6.

fills in the pore space more effectively and the water cannot percolate as far into the snow during the summer. This means that it is closer to the cold surface in the winter and can refreeze. In contrast, the reason a perennial aquifer is sustained in Figure 6II is because the water penetrates sufficiently far that it is insulated from the cold surface.

Above a critical $\overline{Q}$ there is too much melting for the firn to accommodate and runoff begins. The transition from an ac-
cumulation area to an ablation area occurs when runoff exceeds the accumulation. Figure 6IV shows an ablation area where the surface meltwater is only able to enter a few meters into the snow and reaches the impermeable barrier of the glacial ice surface. During the course of the summer all of the snow is melted as well as some of the glacial ice. The streamlines show net upward motion in this case indicating that there is net ablation over the course of the year.

In Figure 7 we examine how mean properties of the firn change as the mean surface forcing varies, for three different values
of accumulation rate. Each point in this figure corresponds to an annual average of a periodic simulation such as those in Figure 6. We quantify the lower firn warming by calculating the mean temperature at the bottom of the domain $\overline{T}_\infty$ and the mean surface temperature $\overline{T}_s$. We also calculate the total quantity of surface melt and the partitioning of the melt between runoff, liquid storage in the ice, and refreezing in the firn. Runoff and melt are calculated from the model output, liquid storage is taken to be the total water flux out of the bottom of the domain and the amount of refreezing is computed as the residual.

For $\overline{Q} < -Q_0$ no melting occurs and the domain top and bottom temperatures are identical. However, as soon as the annual mean surface forcing increases above $-Q_0$, the domain top and bottom temperatures diverge, due to the release of latent heat which warms the snow. Depending on the accumulation rate, the average bottom firn temperature can reach the melting point, corresponding to a perennial firn aquifer. This does not occur for smaller accumulation rates, i.e. figure 7(a), but does for larger accumulation rates, i.e. figure 7(b) and figure 7(c). Additionally, all three panels show that when $\overline{Q}$ increases further
the bottom firn temperature decreases again. This corresponds to the second type of percolation zone shown in figure 6III, in which water only penetrates part of the way into the domain before refreezing. When $\overline{Q}$ is large enough such that the region has become an ablation zone, the bottom temperature (now the temperature of incoming glacial ice) is almost the same as the





surface temperature. The largest firn temperatures occur at intermediate values of surface forcing, considerably lower than the value required to transition to an ablation region.

The thermal structure and water content of the lower firn are strongly tied to the amount of meltwater produced, which in this model is tied directly to the annual mean surface forcing. In a warming world, one can imagine a particular location

transitioning from an accumulation to percolation and then ablation region. Our results show that storage and refreezing can accommodate much of the melt that occurs when the warming is not too large. Once the forcing is sufficient for runoff to start, the amount of refreezing decreases slightly so that an increasingly large fraction of the melt runs off. Most of this runoff is presumably routed to the glacier bed and then the ocean. As well as a form of mass loss, the timing and quantity of meltwater delivery to the bed will determine the style of subglacial drainage system that develops and the subsequent ice dynamics

(Zwally et al., 2002; Schoof, 2010; Tedstone et al., 2015).

## 5  Conclusions

We have described a continuum model for the evolution of firn hydrology, compaction, and themodynamics. The model is capable of determining the evolution of the firn including the temperature, porosity, and water content. The model differs from other models of firn hydrology in its treatment of the percolation of water, for which we use Darcy's law and a parametrization

of capillary pressure. Our treatment for runoff also differs in that we assume that water runs off when the surface layer of snow is fully saturated rather than assuming runoff at depth when the percolating water first reaches an impermeable ice layer.

The model applies to all types of glacier surface, i.e. accumulation, percolation, and ablation areas. Given the forcing (energy flux and accumulation rate), the model selects which of these applies to any particular region. One of the useful outputs of the model is an indication of how the firn may change as function of climate warming, as revealed by moving from left to right in

figure 7. In agreement with Kuipers Munneke et al. (2014) and Steger et al. (2017a) we find that perennial firn aquifers occur when there is sufficiently high accumulation and sufficient melting occurs.

One of the advantages of the continuum formulation is to clarify how mechanical compaction and refreezing combine to determine the evolution of snow density (porosity in our model). Here we have assumed that these processes can be added together, i.e. equation (9). It is quite likely, however, the presence of liquid water affects the mechanical compaction process,

and an improved parametrization of wet compaction could be incorporated within our framework.

In the future, we hope to extend this work beyond the one-dimensional solutions presented here. In principle the model applies to fully three-dimensional geometries, when the slope of the surface will allow meltwater to percolate laterally as well as vertically. The data from Humphrey et al. (2012) suggest the occurrence of 'piping events' where meltwater forms a vertical channel and breaks through to depths where the snow is much colder. These events could be captured in a two-dimensional

framework, and it is possible that a theory allowing the solid ice and liquid water to have different temperatures may help explain these features. On a larger scale, the horizontal scales of the ice sheet are much larger than the depth of the firn, so a reduced, vertically-integrated version of this theory may also be useful.





The use of Darcy's law requires an estimate for the permeability and the relative permeability. The comparison of our model behavior with the data from Humphrey et al. (2012) in figure 4 is encouraging and suggests that these parameters could be determined with detailed measurements of surface melt and snow temperatures. Here we have interpreted the porosity and the permeability as grain scale properties. An alternative interpretation that might be appropriate on larger scales would treat these

as averages over fractures, pipes, and ice lenses, to give a macroscopic effective porosity and permeability.

Although we have focused on idealized, periodic simulations, the model can be forced by real climatological data or coupled to a regional atmospheric model. The model could also be coupled to an ice sheet model, using the deep firn temperature $\overline{T}_\infty$ as the surface boundary condition for the ice sheet.

## Appendix A:  Surface energy balance

The surface energy balance is given by

$$-\overline{K}\frac{\partial T}{\partial z} = -(1-\alpha)S_w - L_w + \epsilon\sigma T^4 - \chi(T_a - T) - \rho_w c_i a(T_a - T) - \rho_w c_w R(T_a - T) + \rho_w \mathscr{L} M, \tag{A1}$$

where the terms represent, in order, conduction into the ice, incoming shortwave radiation $S_w$ ($\alpha$ is the albedo), incoming longwave radiation $L_w$, outgoing longwave radiation ($\epsilon$ is the emissivity and $\sigma$ is the Stefan-Boltzmann constant), turbulent heat transfer with coefficient $\chi$, sensible heat fluxes associated with solid and liquid precipitation, which is assumed to fall with

the air temperature $T_a$, and latent heat flux associated with melting.

Linearizing this equation around the melting temperature $T_m$ gives equation (15) in the text, where the components of $Q$ are given by

$$Q(t) = (1-\alpha)S_w + L_w - \epsilon\sigma T_m^4 + \chi(T_a - T_m) + \rho_w c_i a(T_a - T_m) + \rho_w c_w R(T_a - T_m), \tag{A2}$$

and the effective heat transfer coefficient $h$ includes contributions from turbulent heat transfer and outgoing longwave radiation

as

$$h = \chi + 4\epsilon\sigma T_m^3. \tag{A3}$$

Using the values shown in tables 1 and 2, we determine that a reasonable scale for $Q$ is $Q_0 = 200$ W m$^{-2}$ and $h = 14.8$ W m$^{-2}$ K$^{-1}$.

## Appendix B:  Nondimensional model

We nondimensionalize the lengths by $\ell = Q_0 t_0/(\rho\mathscr{L})$ and time by the annual period $t_0$. We write $T = T_m + \Delta T\theta$ and choose the temperature scale as $\Delta T = Q_0/h$. Enthalpy is scaled with $\rho_i c_i \Delta T$, ice velocity with $\ell/t_0$, water velocity with $(\rho_w g k_0)/\mu$, and water pressure with $\rho_w g\ell$. We define the parameters

$$\mathcal{U} = \frac{\rho g k_0 t_0}{\ell\mu}, \quad \mathcal{S} = \frac{\mathscr{L}}{c_p\Delta T}, \quad Pe = \frac{\rho c_p\ell^2}{Kt_0}, \quad B = \frac{\rho g d_p\ell}{\gamma}. \tag{B1}$$





| | | |
|---|---|---|
| $S_w$ | 292 W m$^{-2}$ | Net shortwave radiation |
| $\alpha$ | 0.6 | Ice albedo |
| $\epsilon$ | 0.97 | Emissivity |
| $\sigma$ | 5.7×10$^{-8}$ W m$^{-2}$ K$^{-4}$ | Stefan-Boltzmann constant |
| $L_w$ | 279 W m$^{-2}$ | Longwave radiation |
| $\chi$ | 10.3 W m$^{-2}$ K$^{-1}$ | Turbulent transfer coefficient |
| $a_0$ | 9.5×10$^{-9}$ m s$^{-1}$ | Accumulation |
| $T_a$ | 267 K | Average air temperature |

**Table 2.** Typical numerical values for the surface energy balance (Cuffey and Paterson, 2010; van den Broeke et al., 2011).

where $\mathcal{U}$ is the scale for the water percolation relative to ice motion, $\mathcal{S}$ is the Stefan number, $Pe$ is the Péclet number and $B$ is the Bond number. Typical parameter values are shown in table 1. Both $\mathcal{U}$ and $\mathcal{B}$ are large; this indicates that the water percolates relatively quickly, and that the percolation is mainly driven by gravity rather than capillary pressure gradients. Both of these could be seen as justification for 'tipping-bucket' type models.

Using the change of variables $Z = z_s(t) - z$, with $\widetilde{w}_i = \dot{z}_s - w_i$, we write the full nondimensional equations as

$$\mathcal{W} = 1 - \phi + \phi S, \tag{B2}$$

$$\mathcal{H} = \mathcal{W}\theta + \mathcal{S}\phi S, \tag{B3}$$

$$\frac{\partial \mathcal{W}}{\partial t} + \frac{\partial}{\partial Z}\left(\widetilde{w}_i \mathcal{W} + q\right) = 0, \tag{B4}$$

$$\frac{\partial \mathcal{H}}{\partial t} + \frac{\partial}{\partial Z}\left[\widetilde{w}_i \mathcal{H} + q\left(\theta + \mathcal{S}\right) - \frac{\mathcal{W}}{Pe}\frac{\partial \theta}{\partial Z}\right] = 0, \tag{B5}$$

$$(1 - \phi)\frac{d\widetilde{w}_i}{dZ} = -c\phi, \tag{B6}$$

$$q = \mathcal{U}k(\phi)k_r(S)\left(1 - \frac{\partial p_w}{\partial Z}\right), \tag{B7}$$

$$p_w = -\frac{1}{B}S^{-\alpha} \quad (S < 1) \quad \text{or} \quad p_w \geq -\frac{1}{B} \quad (S = 1), \tag{B8}$$

subject to the boundary conditions

$$\widetilde{w}_i \mathcal{H} + q\left(\theta + \mathcal{S}\right) - \frac{1}{Pe}\mathcal{W}\frac{\partial \theta}{\partial Z} = \mathcal{S}\left[Q - \theta + R - r\right] \quad \text{on} \quad Z = 0, \tag{B9}$$

$$\widetilde{w}_i \mathcal{W} + q\left(\theta + \mathcal{S}\right) = a + R - r \quad \text{on} \quad Z = 0, \tag{B10}$$

$$\widetilde{w}_i = \frac{a - M}{1 - \phi_0} \quad (\widetilde{w}_i > 0) \quad \text{or} \quad \widetilde{w}_i = \frac{a - M}{1 - \phi} \quad (\widetilde{w}_i \leq 0) \quad \text{on} \quad Z = 0, \tag{B11}$$

$$-\mathcal{U}k(\phi)k_r(S)\frac{\partial p_w}{\partial Z}\left(\theta + \mathcal{S}\right) \rightarrow 0 \quad \text{as} \quad Z \rightarrow \infty, \tag{B12}$$

$$-\frac{1}{Pe}\mathcal{W}\frac{\partial \theta}{\partial Z} \rightarrow 0 \quad \text{as} \quad Z \rightarrow \infty. \tag{B13}$$

$$\tag{B14}$$





## Appendix C: Refreezing front

Here we detail the approximate solution for the refreezing front considered in section 3.1. The schematic is shown in figure 2(a). We use dimensionless variables and the equations we solve are

$$\phi\frac{\partial S}{\partial t} + \frac{\partial q}{\partial Z} = 0, \qquad (0 < Z < Z_f) \tag{C1}$$

$$q = \mathcal{U}k(\phi)k_r(S)\left(1 + \frac{1}{B}p_c'(S)\frac{\partial S}{\partial Z}\right), \qquad (0 < Z < Z_f) \tag{C2}$$

$$\frac{\partial\phi}{\partial t} = 0, \qquad (0 < Z < Z_f) \quad \text{and} \quad (Z > Z_f) \tag{C3}$$

$$\frac{\partial\theta}{\partial t} = \frac{1}{Pe}\frac{\partial^2\theta}{\partial Z^2}, \qquad (Z > Z_f) \tag{C4}$$

The boundary conditions for equations (C1) to (C4) are

$$\theta = \theta_\infty \qquad \text{as} \qquad Z \to \infty, \tag{C5}$$

$$\theta = 0, \, S = 0 \qquad \text{on} \qquad Z = Z_f. \tag{C6}$$

$$q = R \qquad \text{on} \qquad Z = 0, \tag{C7}$$

where $\theta_\infty < 0$ is the cold far-field temperature, and $R$ is the prescribed constant rainfall rate. Integrating across the front at $Z_f(t)$ gives the nondimensional jump conditions

$$\left[q + \phi S\left(\tilde{w}_i - \dot{Z}_f\right)\right]_-^+ = -m_I, \tag{C8}$$

$$\left[(1 - \phi)\left(\tilde{w}_i - \dot{Z}_f\right)\right]_-^+ = m_I, \tag{C9}$$

$$\frac{1}{Pe}\left[(1 - \phi)\frac{\partial\theta}{\partial Z}\right]_-^+ = -\mathcal{S}m_I, \tag{C10}$$

which states that the mass $-m_I$ that freezes from the liquid phase enters the solid phase, and that the latent heat from refreezing warms the dry ice below. We can simplify these equations since $\theta = 0$ in the upper portion $(+)$, $\phi = \phi_0$ and $S = 0$ in the lower portion $(-)$, and the ice velocity $\tilde{w}_i$ is zero, so

$$q - \phi^+ S^+ \dot{Z}_f = -m_I, \tag{C11}$$

$$(\phi^+ - \phi_0)\dot{Z}_f = m_I, \tag{C12}$$

$$\frac{1}{Pe}(1 - \phi_0)\left.\frac{\partial\theta}{\partial Z}\right|_- = \mathcal{S}m_I, \tag{C13}$$

After a short initial transient, the solution approximates a travelling wave in which the upper region $0 < Z < Z_f$ has $\theta = 0$, $\phi = \phi^+$ (to be determined shortly), and $q = R$. Since $B \gg 1$, this means $\mathcal{U}k(\phi^+)k_r(S^+) \approx R$, which determines the constant $S_+$ in the upper region (there is a narrow boundary layer behind the front, in which $S_+$ changes rapidly but $q - \phi^+ S^+ \dot{Z}_f$ is constant; see below).

We next solve for the temperature evolution in the lower region. Assuming that the freezing front moves quickly, i.e. $|\dot{Z}_f| \gg 1$ (this is appropriate since $\mathcal{U}$ is large), we can move into a translating frame $\tilde{Z} = Z - Z_f$ and neglect the time dependence so



that

$$\frac{1}{Pe}\frac{\partial \theta}{\partial \tilde{Z}} + \dot{Z}_f \theta \approx \dot{Z}_f \theta_\infty, \tag{C14}$$

is constant (set by the far-field temperature), and hence $\theta \approx \theta_\infty \left(1 - e^{-Pe\dot{Z}_f \tilde{Z}}\right)$. This is the approximate solution given dimensionally in equation (39). From the temperature field we can determine the melt rate using equation (C13) as

$$m_I = (1 - \phi_0)\frac{\dot{Z}_f \theta_\infty}{\mathcal{S}}, \tag{C15}$$

which is negative, corresponding to freezing, since $\theta_\infty < 0$, and equation (C12) therefore determines the porosity jump,

$$\phi^+ = \phi_0 + (1 - \phi_0)\frac{\theta_\infty}{\mathcal{S}}. \tag{C16}$$

Finally, the jump condition for water conservation, equation (C11), determines the speed of the front as

$$\dot{Z}_f = \frac{R\mathcal{S}}{\phi^+ S^+ \mathcal{S} - (1 - \phi_0)\theta_\infty}. \tag{C17}$$

This result corroborates the front velocity derived by Colbeck (1972), Gray (1996), and Durey (2014).

To capture the smoothing of the front due to capillary pressure, we can examine the narrow boundary layer behind the front. The relevant scale for this region is of order $1/B$, so we write $Z - Z_f = \hat{Z}/B$, and determine the leading-order quasi-static approximation

$$-\phi\dot{Z}_f\frac{\partial S}{\partial \hat{Z}} + \mathcal{U}\frac{\partial}{\partial \hat{Z}}\left[k(\phi)k_r(S)\left(p_c'(S)\frac{\partial S}{\partial \hat{Z}} + 1\right)\right] = 0, \tag{C18}$$

with the boundary conditions

$$S \to S^+ \quad \text{as} \quad \hat{Z} \to -\infty \quad \text{and} \quad S = 0 \quad \text{on} \quad \eta = 0. \tag{C19}$$

We can integrate this once and find

$$\mathcal{U}k(\phi)k_r(S) - \phi S\dot{Z}_f + \mathcal{U}k(\phi)k_r(S)p_c'(S)\frac{\partial S}{\partial \hat{Z}} = \mathcal{U}k(\phi)k_r(S_+) - \phi^+ S^+ \dot{Z}_f, \tag{C20}$$

where the constant comes from the matching condition. If we now make use of $p_c = S^{-\alpha}$, $k_r = S^\beta$ and take $\beta = 2$, $\alpha = 1$, then equation (C20) becomes

$$\frac{\partial S}{\partial \hat{Z}} = S^2 - S_+^2 - \psi(S - S_+), \tag{C21}$$

where $\psi = \frac{\phi\dot{Z}_f}{\mathcal{U}k(\phi)}$. which can be integrated to give

$$S = \frac{\psi}{2} + \frac{2S^+ - \psi}{2}\tanh\left\{\text{arctanh}\left(\frac{\psi}{\psi - 2S^+}\right) - \frac{2S^+ - \psi}{2}\hat{Z}\right\}, \tag{C22}$$

which is similar to the result derived by Gray (1996).





## Appendix D: Saturation fronts

Here we calculate the motion of the fully saturated fronts for isothermal conditions with fixed porosity $\phi = \phi_0 e^{-Z/Z_0}$, as in section 3.3. We again make use of dimensionless variables. In the time before full saturation initiates, and in the limit $B \gg 1$, conservation of water at the wetting front $Z_f(t)$ is given, as in the appendix C with $\theta_\infty = 0$, by

$$R - \phi_f S_f \dot{Z}_f = 0, \tag{D1}$$

where $\phi_f(t) = \phi_0 e^{-Z_f/Z_0}$ is the porosity at the front and $S_f$ is the saturation. Using permeability $k(\phi) = \phi^3$ and relative permeability $k_r(S) = S^2$, we can calculate the saturation induced by the rain fall as

$$S_f = \left( \frac{R}{\mathcal{U}\phi_f^3} \right)^{1/2}. \tag{D2}$$

Thus, the initial evolution equation for the front before full saturation is

$$\dot{Z}_f = \sqrt{\mathcal{U}R\phi_0} \exp\left\{ -\frac{Z_f}{2Z_0} \right\}, \tag{D3}$$

which can be integrated to give

$$Z_f = 2Z_0 \ln\left\{ 1 + \frac{\sqrt{\mathcal{U}R\phi_0}}{2Z_0} t \right\}. \tag{D4}$$

We can therefore calculate the position of the front, and the time, at which full saturation occurs by setting $S_f = 1$. This gives

$$Z_1 = \frac{Z_0}{3} \ln\left\{ \frac{\phi_0^3 \mathcal{U}}{R} \right\} \quad \text{and} \quad t_1 = \frac{2Z_0}{\sqrt{\mathcal{U}R\phi_0}} \left[ \left( \frac{\phi_0^3 \mathcal{U}}{R} \right)^{1/6} - 1 \right]. \tag{D5}$$

Now in the fully saturated region, between the upper and lower saturation fronts $Z_u(t) < Z < Z_l(t)$, we have that

$$\mathcal{U}k(\phi)\left( 1 - \frac{\partial p_w}{\partial Z} \right) = q_s, \tag{D6}$$

where $q_s$ is the water flux in the fully saturated region, which is constant since there is no compaction. Rearranging and integrating again, using $p_w(Z_u) = p_w(Z_l)$, gives

$$Z_l - Z_u = \frac{q_s}{\mathcal{U}} \int_{Z_u}^{Z_l} \frac{dy}{k(\phi)}, \tag{D7}$$

which determines the flux as

$$q_s = \frac{3\phi_0^3 \mathcal{U}(Z_l - Z_u)}{Z_0 \left[ e^{3Z_l/Z_0} - e^{3Z_u/Z_0} \right]}. \tag{D8}$$

Since there is no melting/refreezing, water conservation across the lower front states that

$$q_s - \phi_l \dot{Z}_l = 0. \tag{D9}$$





The equivalent jump condition on the upper front is

$$q_s - \phi_u \dot{Z}_u = R - \phi_u S_u \dot{Z}_u, \tag{D10}$$

where $S_u = (R/\mathcal{U}\phi_u^3)^{1/2}$ as before. Thus, once full saturation is initiated, we must solve the ODEs

$$\dot{Z}_l = \frac{q_s}{\phi_l} \quad \text{and} \quad \dot{Z}_u = \frac{q_s - R}{\phi_u \left[ 1 - \left( \frac{R}{\mathcal{U}\phi_u^3} \right)^{1/2} \right]} \quad \text{with} \quad q_s = \frac{3\phi_0^3 \mathcal{U}(Z_l - Z_u)}{Z_0 \left[ e^{3Z_l/Z_0} - e^{3Z_u/Z_0} \right]}, \tag{D11}$$

5    subject to the initial conditions

$$Z_u = Z_l = Z_1 \quad \text{at time} \quad t = t_1. \tag{D12}$$

In dimensional form, these are the same as equation (41), and the solutions are compared to the full numerical solution using the enthalpy method in figure 5.

*Acknowledgements.*  We wish to thank the 2016 Geophysical Fluid Dynamics summer program at the Woods Hole Oceanographic Institution,
10   which is supported by the National Science Foundation and the Office of Naval Research. We also acknowledge financial support by NSF
grant DGE1144152 (CRM) as well as Marie Curie FP7 Career Integration Grant within the 7th European Union Framework Programme.
(IJH).





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
