# Peer review of "A continuum model for meltwater flow through compacting snow"

_The Cryosphere, 2017_

## Referee Comment (RC1) · Anonymous Referee #1 · 31 Aug 2017

The manuscript presents a one-dimensional continuum model describing the flow of meltwater in a compacting snowpack. Compaction of dry snowpacks, and meltwater flow through non-compacting snowpacks had been considered in the literature before separately, but the coupling between the two hasn't been addressed, as far as I know. This latter coupling is key to examine the dynamics of meltwater percolating in thick firn layers, such as those present in Greenland, and to develop an understanding of how these layers respond to a warming climate. The authors present a mathematical model for these coupled dynamics, for which they derive both analytical and numerical solutions. They identify four different scenarios depending on the intensity of the surface energy forcing, and also analyze how the transition among those scenarios is affected by changes in annual surface accumulation. An important result is that firn

can switch from a sponge-like behavior, such that all the melt is stored within the firn, to the behaviour of an almost impermeable substrate that allows further meltwater to run off the ice sheet. Interestingly, saturation of the whole layer is not required to start surface runoff.

Most of my comments below are relatively minor. The main thing I would like to see is more focus on the physical processes that sit behind the response of firn to changes in the surface forcing (more detailed comments in this respect in the minor comments below). In this respect, I wonder whether the two simplified problems discussed in Sec. 3 could also be used to illustrate some of the fundamental aspects of the flow of meltwater through firn in isolation, thus helping to explain the physical basis of the more complex results presented in Sec.4. In particular, whether saturation is achieved or not seems to be key to understanding the two different types of percolation zones, and hence whether perennial water storage is possible in the firn layer. In my view it would be useful if the authors could place more constraints on the physics that controls this switch, which seems to be the most interesting result in the paper. Last, I think that a slightly more critical literature review in the introduction, in particular when it comes to comparing and contrasting previous approaches to the present work, would also be helpful.

Minor comments:

Introduction page 2, line 26: mention that the test problems in Sec. 3 are primarily used to benchmark the numerics. Also, I agree that the results of these test problems compare favorably with data, but I'd like to see a mention to the fact that no mechanical compaction is considered in obtaining such results.

Section 2

- eqs 1-3: the velocity vector u is undefined

- page 5, paragraph 2.2: I am not an expert on the subject, but I wonder whether the

specific choice of the parameterization for the compaction rate produces any qualitative difference in the results. I would like to see more discussion on this.

- eq 11: $w_i$ is undefined

- eq 24: not sure gamma is defined. Surface tension?

Section 3

General comments, expanding on my major points above:

- Sec 3.1: You demonstrate that refreezing happens at the front, and so porosity behind the front decreases. Can you form ice lenses with this mechanism, once you allow for a depth-dependent porosity profile? Along similar lines, on page 10, line 10, you state that the effect of refreezing is to slow the propagation of the front and to increase porosity as the front passes. How would this result change if you allowed for compaction/ a depth-dependent porosity profile? Does this have anything to do with the switch between the two qualitatively different percolation zones of fig. 6II and 6III? See next comment for a related point

- Sec. 3.2: I agree with the authors that the agreement between theory and observations is good. It seems to me though (considering your expression for $\dot{Z}_f$, eq. 38) that the necessity to fit two different front velocities highlights once more that propagation of a water front in cold snow is strongly affected by compaction/ the porosity stratification. I'd like to see more discussion on this.

Minor comments:

- line 19: replace understand with understood

- Figure 2: the color scheme is not explained. Panel a: the text $T=T_m$ should not be placed below the front, it's confusing. Maybe on the side of $\dot{Z}_f$?

- eq. 28: what does the ' stand for?

Section 4

- Pages 13-15: The explanation of the differences between the two percolation zones is very qualitative and a little bit vague as a result, in my opinion. It seems to me that the key point is that in one case (fig 6II) there is unsaturated flow, whereas in the other case (6III) saturation is attained. Your results seem to suggest that a saturated from propagates more slowly than an unsaturated one, and hence penetrates less in depth preventing the formation of the firn aquifer. Would you be able to comment on this?

---

## Referee Comment (RC2) · Anonymous Referee #2 · 23 Sep 2017

Overall, I think this paper and modeling effort is an important advancement in the modeling meltwater movement through firn in that it addresses the more complex physics of energy and fluid flow beyond the bucket tipping methods.  It also combines compaction and fluid flow.  It addresses a modeling need that is timely and relevant and the methods and conclusions are valid.

The paper, particularly the introduction and results, are vague and needs significant editing.  In general, the paper would be much improved with specific definitions, for instance, near-surface (>10 m). The Introduction should be rewritten completely and there are suggestion in specific comments.  Additionally, the results need to be rewritten with values given.  The text just mentions the figures and words such as large, small, intermediate.  These need to be defined with numbers and ranges.  This paper and work is significant and worthy of publication after major editing for clarity, adding column headers to all chart, making all figures a simple as possible for the reader to understand, grammar corrections, and defining all acronyms and model variables.  Additionally there are incorrect or vague statements about firn properties and the interaction of meltwater with the subglacial hydrologic system and ice dynamics that should either be removed or clarified.

Specific Comments:

p1, l 2  meltwater can also percolate and store (See work on Greenland firn aquifers) and should be added to this sentence.

p1, l 10 Largest and intermediate are vague.  Please clarify with numbers.

The first two paragraphs in the introduction need major change.  They are vague and contain misstatements.  For instance, while percolated meltwater does affect near-surface firn temperatures (~ 1 m depth), the surface temperature of a glacier is mainly atmospherically driving by conduction (top 10's of cm).  Be clear and precise in wording.  Also it the first paragraph makes the assumption that all meltwater that reaches the bed causes a dynamical response in the ice which is incorrect.  Define the depth for mean firn temperature, near-surface and relatively cold. The paragraphs should also include citations including work by Fountain, Harper,  Humphreys, Koenig with specific numbers on the buffering potential of the firn and how much of the buffering potential is likely filled already.

p2, l 3 IMAU-FDM is not defined.

p2, l 8: The SNOWPACK model now includes Richards equations to describe fluid flow in variably saturated media. How does this model compare to that?  See Wever, N., et al. "Solving Richards Equation for snow improves snowpack meltwater runoff estimations in detailed multi-layer snowpack model." *The Cryosphere* 8.1 (2014): 257-274.  Also SNOWPACK should be cited.

The last 4 paragraphs in the Introduction should not be included in an introduction. They include topics of methods, discussion, etc. The introduction should just discuss what is need to set up your scientific questions. More the rest to the proper sections. It is OK to have 3-4 sentences outlining your paper at the end of the introduction but this is too much and is confusing for the reader.

There are some variables not defined in equations:
eq (1) - u
eq. (12) cp, K

p 3, l 23, can they explain why they selected the Carman-Kozeny relationship?

P3, l25 Generally a table is introduced in a scientific paper with a sentence similar to Table 1 provides the ….. Please change to this format. See table 1 for the parameter values we use later is vague. Also Table 1 should have column headers such as parameter and value.

p 4, figure 1 caption: maybe change "squeezed out" to "replaced by water" . The final sentence "Ice grains make contact in the third dimension and similarly many of the air and water pockets are connected in the third dimension." is unclear. How is this picture showing a third dimension?

p 5, l 6: they assume that compaction is unaltered by metlwater. Can they justify that assumption or at least describe how compaction actually is affected by meltwater and what that might do to their results.

P 5, l 9 State why Herron and Langway was chosen over the other compaction equations, especially since it was developed for dry snow where Morris and Wingham had more variety in location.

p 8, l 19: change "understand" to "understood"

P 9 figure 2, While this appears correct it could benefit if you replaces some variables with terms like rain (R), Saturation (s). Also the lines and colors do not appear to be explained.

P10 figure 3 Similar comments to figure 2. If you can label the y axis with rate of refreeze it would be helpful. The figures are difficult to understand.

P13 l 4 define small surface layer and full firn column with approximate depths.

P 13, l 30 only the Forster citation is appropriate here. The others are studies from the West Coast of Greenland where water may persist late into the season but has not been confirmed to be perennial except in buried lakes.

p 13, l 32: I think they mean percolation instead of accumulation? The explanation of this scenario doesn't make sense to me (e.g., that the meltwater fills in the pore space more

effectively and that prevents water from moving deeper (p 15, l 1). Are there field observations to justify this? Increased water saturation (more effective pore space filling) results in higher hydraulic conductivity, which would allow more water to move through the column. Why doesn't the water move deeper? Is it running off at the surface? If so, that should be clarified. This may also be a result of this being a 1D model - in 2D the water would be able to flow laterally and you might not get this result.

Could they detail more about the relative sensitivity to accumulation vs surface energy balance?

P 15, l 3 Cite here that your model is consistent with others, Kuipers Munneke for instance.

P15 Throughout this entire page it is unclear what values are being refered to.  For instance they mention the critical Q value, Q being large enough.  Specify with numbers.  Also there are methods in the results for instance  "We also calculate the total quantity of surface melt and the partitioning of the melt between runoff, liquid storage in the ice, and refreezing in the firn. Runoff and melt are calculated from the model output, liquid storage is taken to be the total water flux out of the bottom of the domain and the amount of refreezing is computed as the residual"  Move this to the methods section and just report results here.

p 15, l 14: why is storage equal to the flux out? Shouldn't storage be what doesn't leave the system?

P16 l 17  Glacier surface should be Glacier facies,  Make sure this is consistent throughout.

p 16, l 27: can they define what surface slope  they are referring to that controls  "lateral percolation"?  Also, I would say lateral flow.

---

## Author Comment (AC1) · 13 Oct 2017

The comment was uploaded in the form of a supplement:
https://www.the-cryosphere-discuss.net/tc-2017-128/tc-2017-128-AC1-supplement.pdf

---

## Author Response (AR1)

**Response to referees**

Colin R. Meyer and Ian J. Hewitt

October 13, 2017

We wish to thank both referees for thoughtful and helpful reviews. Throughout, the original comments are in black text and our responses are colored in blue text.

**Comments to referee 1**

**Major Comments**

The manuscript presents a one-dimensional continuum model describing the flow of meltwater in a compacting snowpack. Compaction of dry snowpacks, and meltwater flow through non-compacting snowpacks had been considered in the literature before separately, but the coupling between the two hasn't been addressed, as far as I know. This latter coupling is key to examine the dynamics of meltwater percolating in thick firn layers, such as those present in Greenland, and to develop an understanding of how these layers respond to a warming climate. The authors present a mathematical model for these coupled dynamics, for which they derive both analytical and numerical solutions. They identify four different scenarios depending on the intensity of the surface energy forcing, and also analyze how the transition among those scenarios is affected by changes in annual surface accumulation. An important result is that firn can switch from a sponge-like behavior, such that all the melt is stored within the firn, to the behaviour of an almost impermeable substrate that allows further meltwater to run off the ice sheet. Interestingly, saturation of the whole layer is not required to start surface runoff.

Most of my comments below are relatively minor. The main thing I would like to see is more focus on the physical processes that sit behind the response of firn to changes in the surface forcing (more detailed comments in this respect in the minor comments below). In this respect, I wonder whether the two simplified problems discussed in Sec. 3 could also be used to illustrate some of the fundamental aspects of the flow of meltwater through firn in isolation, thus helping to explain the physical basis of the more complex results presented in Sec.4. In particular, whether saturation is achieved or not seems to be key to understanding the two different types of percolation zones, and hence whether perennial water storage is possible in the firn layer. In my view it would be useful if the authors could place more constraints on the physics that controls this switch, which seems to be the most interesting result in the paper. Last, I think that a slightly more critical literature review in the introduction, in particular when it comes to comparing and contrasting previous approaches to the present work, would also be helpful.

Thanks for highlighting the most interesting aspects of the paper and outlining these shortcomings. The two benchmark problems are designed to show the transition between

saturated and unsaturated regions as well as saturated and fully saturated regions. Both of these behavior become important in the transition between an accumulation zone with and without a perennial firn aquifer. Both referees identified this switch in behaviour between figure 6II and III as being mysterious; we have rewritten the discussion around this transition which we hope will make the physics clearer (see below). We have rewritten much of the introduction in response to the other referee, and included more references that compare and contrast previous approaches.

**Minor Comments**

1. Introduction page 2, line 26: mention that the test problems in Sec. 3 are primarily used to benchmark the numerics. Also, I agree that the results of these test problems compare favorably with data, but I'd like to see a mention to the fact that no mechanical compaction is considered in obtaining such results.

   We now explicitly state that there is no mechanical compaction in either test problem.

2. Section 2
   - eqs 1-3: the velocity vector u is undefined
   - page 5, paragraph 2.2: I am not an expert on the subject, but I wonder whether the specific choice of the parameterization for the compaction rate produces any qualitative difference in the results. I would like to see more discussion on this.
   - eq 11: $w_i$ is undefined
   - eq 24: not sure gamma is defined. Surface tension?

   Thanks for pointing out these various oversights. We have added the definitions (surface tension is defined where it is introduced following equation (6)). In our preparation we used several parametrizations for compaction and our code is general enough to accept other forms of the compaction law. However, from the experimenting we have done, the different compaction relations produce quantitative rather than qualitative changes to our results and so we have adopted the Herron and Langway relation for simplicity and to facilitate comparisons with other papers. We have added a statement to the text saying that changing the compaction parameterization does not qualitatively change our results.

3. Section 3
   General comments, expanding on my major points above:
   - Sec 3.1: You demonstrate that refreezing happens at the front, and so porosity behind the front decreases. Can you form ice lenses with this mechanism, once you allow for a depth-dependent porosity profile? Along similar lines, on page 10, line 10, you state that the effect of refreezing is to slow the propagation of the front and to increase porosity as the front passes. How would this result change if you allowed for compaction/ a depth-dependent porosity profile? Does this have anything to do with the switch between the two qualitatively different percolation zones of fig. 6II and 6III? See next comment for a related point

   This a great point. Ice lenses can form if the pre-existing porosity is small enough. In that case, a a saturated region will form above the (impermeable) lens. We have added a comment to the text about this.

4. - Sec. 3.2: I agree with the authors that the agreement between theory and observations is good. It seems to me though (considering your expression for $\dot{Z}_f$, eq. 38) that the necessity to fit two different front velocities highlights once more that propagation of a water front in cold snow is strongly affected by compaction/ the porosity stratification. I'd like to see more discussion on this.

   Thanks to the referee for engaging with this section comparing theory and data. There are a number of reasons why the front speed may vary with depth; one of these is the porosity stratification, as the reviewer notes. However, given the parameters in our model, a decrease in porosity with depth actually leads to a decrease in front speed (because the decrease in porosity results in an increase in saturation that means the average speed of the percolating water decreases). The greater control on the speed of the front is the surface melt rate that provides the flux from the surface. Our analysis suggests that the increase in front speed most likely reflects an increase in the melt inflow from the surface. We added a statement to this section to comment that we attribute the increase in speed to the surface melt rate.

5. Minor comments:
   - line 19: replace understand with understood
   - Figure 2: the color scheme is not explained. Panel a: the text $T = T_m$ should not be placed below the front, it's confusing. Maybe on the side of $\dot{Z}_f$? - eq. 28: what does the ' stand for?

   Thanks for spotting these omissions. We have fixed the grammar, described the schematic colors, and placed $T = T_m$ above the front. The ' symbol stands for the derivative with respect to $S$ and we have added this to the text.

6. Section 4
   - Pages 13-15: The explanation of the differences between the two percolation zones is very qualitative and a little bit vague as a result, in my opinion. It seems to me that the key point is that in one case (fig 6II) there is unsaturated flow, whereas in the other case (6III) saturation is attained. Your results seem to suggest that a saturated from propagates more slowly than an unsaturated one, and hence penetrates less in depth preventing the formation of the firn aquifer. Would you be able to comment on this?

   We have revised this section in order to better discuss the physics governing this transition. Our results indicate that the snow becomes saturated in case III because the porosity decreases more rapidly with depth, and the water is therefore not able to percolate deeper into the snowpack (the permeability is lower and the amount of space for it to occupy at depth is less). With more water entering the snowpack and refreezing (than in case II), the pore space fills in with refrozen water (this is the reason why the porosity decreases more rapidly with depth in case III). Consequently the summer saturations are larger and the meltwater is stored closer to the surface, which therefore means that it can all refreeze in the winter. There is therefore a link between full saturation being achieved and the lack of a perennial aquifer; but the model suggests this is driven the other way to that suggested by the reviewer. *i.e.* there is not a fundamental change in percolation rate that occurs when full saturation is achieved, but rather the increased saturation reflects the reduction in porosity that is responsible for slowing down/limiting the depth of percolation.

**Comments to referee 2**

**Major Comments**

Overall, I think this paper and modeling effort is an important advancement in the modeling meltwater movement through firn in that it addresses the more complex physics of energy and fluid flow beyond the bucket tipping methods. It also combines compaction and fluid flow. It addresses a modeling need that is timely and relevant and the methods and conclusions are valid.

The paper, particularly the introduction and results, are vague and needs significant editing. In general, the paper would be much improved with specific definitions, for instance, near-surface (>10 m). The Introduction should be rewritten completely and there are suggestion in specific comments. Additionally, the results need to be rewritten with values given. The text just mentions the figures and words such as large, small, intermediate. These need to be defined with numbers and ranges. This paper and work is significant and worthy of publication after major editing for clarity, adding column headers to all chart, making all figures a simple as possible for the reader to understand, grammar corrections, and defining all acronyms and model variables. Additionally there are incorrect or vague statements about firn properties and the interaction of meltwater with the subglacial hydrologic system and ice dynamics that should either be removed or clarified.

Thanks to the reviewer for the positive assessment of the paper despite some differences of opinion about style. We have rewritten the introduction to include some of the references suggested below as well as to move some of the paper-outline discussion to later in the paper. We have included more numerical values where appropriate, although we do not think it is always helpful to do so, if the values quoted depend on other parameters that may vary. We have tried to focus on the structure of the solutions rather than the quantitative values, for which particular climatic forcings would need to be taken into account.

**Minor Comments**

1. p1, l 2 meltwater can also percolate and store (See work on Greenland firn aquifers) and should be added to this sentence.

   Agreed, this sentence was loose. We have rewritten several sentences in the abstract and changed the wording so that the percolated water 'may' refreeze.

2. p1, l 10 Largest and intermediate are vague. Please clarify with numbers. The first two paragraphs in the introduction need major change. They are vague and contain misstatements. For instance, while percolated meltwater does affect near-surface firn temperatures ($\sim$1 m depth), the surface temperature of a glacier is mainly atmospherically driving by conduction (top 10's of cm). Be clear and precise in wording. Also it the first paragraph makes the assumption that all meltwater that reaches the bed causes a dynamical response in the ice which is incorrect. Define the depth for mean firn temperature, near-surface and relatively cold. The paragraphs should also include citations including work by Fountain, Harper, Humphreys, Koenig with specific numbers on the buffering potential of the firn and how much of the buffering potential is likely filled already.

   We have rewritten the introduction and clarified the language as suggested. We have

removed some of the vague depth descriptions, but we think that for the most part it is appropriate here to use qualitative descriptions rather than numbers, as the numerical values generally depend on forcing parameters (particularly accumulation rate). We have added comments to the introduction to make clear that our paper is about how the firn column behaves under steady climate conditions; it does not specifically address 'buffering potential', which we consider to be a transient concept. References are given to the other works suggested, which include estimates of this potential.

3. p2, l 3 IMAU-FDM is not defined.

As part of our rewriting of the introduction we now write out Firn Densification Model in full (we consider the expansion of IMAU to be no more transparent, since we only refer to the model in this way to provide consistency with the papers of Steger (2017), who refer to the model as IMAU-FDM).

4. p2, l 8: The SNOWPACK model now includes Richards equations to describe fluid flow in variably saturated media. How does this model compare to that? See Wever, N., et al. "Solving Richards Equation for snow improves snowpack meltwater runoff estimations in detailed multi-layer snowpack model" The Cryosphere 8.1 (2014): 257-274. Also SNOWPACK should be cited.

Many thanks to the reviewer for bringing this paper to our attention. We have added a sentence describing the model comparison in Wever et al (2014) and a citation to Bartlet and Lehning (2002) for SNOWPACK.

5. The last 4 paragraphs in the Introduction should not be included in an introduction. They include topics of methods, discussion, etc. The introduction should just discuss what is need to set up your scientific questions. More the rest to the proper sections. It is OK to have 3-4 sentences outlining your paper at the end of the introduction but this is too much and is confusing for the reader.

We have re-written the introduction and moved some of this discussion to the relevant sections later in the paper.

6. There are some variables not defined in equations: eq (1) - u eq. (12) cp, K

Thanks to the reviewer for catching this oversight.

7. p 3, l 23, can they explain why they selected the Carman-Kozeny relationship?

The Carman-Kozeny is a common relationship between permeability and porosity. Many authors in the snow hydrology literature use Carman-Kozeny and so it facilitates comparisons between studies; we have added a citation to Gray (1996).

8. P3, l25 Generally a table is introduced in a scientific paper with a sentence similar to Table 1 provides the ... Please change to this format. See table 1 for the parameter values we use later is vague. Also Table 1 should have column headers such as parameter and value.

We have followed the reviewer's suggestion and revised the sentence accordingly. We think the contents of the table is self-explanatory without the need for headers.

9. p 4, figure 1 caption: maybe change "squeezed out" to "replaced by water" . The final sentence 'Ice grains make contact in the third dimension and similarly many of the air and water pockets are connected in the third dimension.' is unclear. How is this picture showing a third dimension?

We have changed the wording as suggested. The figure is not intending to show the third dimension (the sentence was included simply to clarify that the ice grains are not levitating as it appears in the 2D cross-section).

10. p 5, l 6: they assume that compaction is unaltered by metlwater. Can they justify that assumption or at least describe how compaction actually is affected by meltwater and what that might do to their results.

This comment highlights an important aspect of our paper. In our model, compaction of porous snow has two parts: mechanical compaction (parametrized by the Herron and Langway model) as well as refreezing by percolating meltwater. We assume that these two processes are independent at a given instance of time and can be added together. The physics of wet snow compaction is sufficiently underdeveloped that we do not know how meltwater may affect the mechanical compaction. One can imagine, for instance, that the presence of thin films of water between ice crystals may affect how fast they are able to slide past one another. We have added to the discussion around this part of the text.

11. P 5, l 9 State why Herron and Langway was chosen over the other compaction equations, especially since it was developed for dry snow where Morris and Wingham had more variety in location.

The Herron and Langway model was chosen for simplicity and consistency with other studies. The Morris and Wingham parameterisation is slightly more complex (with a temperature-history-dependence to one of the coefficients), and may be investigated in future work.

12. p 8, l 19: change "understand" to "understood"

Thanks for catching this.

13. P 9 figure 2, While this appears correct it could benefit if you replaces some variables with terms like rain (R), Saturation (s). Also the lines and colors do not appear to be explained.

We think that the lines across the figure were rendering artefacts. To clarify the information, we have used a new rendering that does not have lines through it, and explain in the caption that the grayscale indicates saturation.

14. P10 figure 3 Similar comments to figure 2. If you can label the y axis with rate of refreeze it would be helpful. The figures are difficult to understand.

We are not sure what is being referred to here. The $y$ axes is the depth below the surface, and the lines show temperature, porosity, and saturation profiles at three different times.

15. P13 l 4 define small surface layer and full firn column with approximate depths.

We now define the small surface layer as $\sim$1m and the full firn column as $\sim$10's m.

16. P 13, l 30 only the Forster citation is appropriate here. The others are studies from the West Coast of Greenland where water may persist late into the season but has not been confirmed to be perennial except in buried lakes.

   Thanks for pointing this out. We have removed the other citations and added a reference to Koenig et al (2014) as well.

17. p 13, l 32: I think they mean percolation instead of accumulation? The explanation of this scenario doesn't make sense to me (e.g., that the meltwater fills in the pore space more effectively and that prevents water from moving deeper (p 15, l 1). Are there field observations to justify this? Increased water saturation (more effective pore space filling) results in higher hydraulic conductivity, which would allow more water to move through the column. Why doesn't the water move deeper? Is it running off at the surface? If so, that should be clarified. This may also be a result of this being a 1D model - in 2D the water would be able to flow laterally and you might not get this result. Could they detail more about the relative sensitivity to accumulation vs surface energy balance?

   As a clarifying measure, we have removed any use of the moniker 'percolation zone' in favor of accumulation and ablation zones. Both reviewers have commented on scenario III and the discussion surrounding it. Our model predicts that despite the increased saturation, the water is not able to move through the column, because it has reached an impermeable barrier below; in fact, the increased saturation is really a *result* of there being reduced porosity (and hence permeability) lower in the firn column which limits how rapidly the water can percolate deeper. However, because there is a larger quantity of water (than in scenario II), when it freezes it fills in the pore space and means that the hydraulic conductivity is lower for the following year. The result is that the porosity decreases more rapidly with depth because it has been filled in by re-frozen water. We agree that things may be different in 2D, although the fact that pore space gets filled with previously frozen water so that there is less space to allow deep percolation is likely still to be the case. We have edited the discussion of this scenario in the text.

18. P 15, l 3 Cite here that your model is consistent with others, Kuipers Munneke.

   We have added a statement in the text to say that our results are consistent with Kuipers Munneke.

19. P15 Throughout this entire page it is unclear what values are being refered to. For instance they mention the critical Q value, Q being large enough. Specify with numbers. Also there are methods in the results for instance "We also calculate the total quantity of surface melt and the partitioning of the melt between runoff, liquid storage in the ice, and refreezing in the firn. Runoff and melt are calculated from the model output, liquid storage is taken to be the total water flux out of the bottom of the domain and the amount of refreezing is computed as the residual" Move this to the methods section and just report results here.

   The specific values that are being referred to here are different for different accumulation rates, and for different amplitude of surface energy forcing. We do not think that augmenting the text with lots of numerical values here is particularly meaningful, but we have added some specific values as examples. The emphasis is on how the

behaviour changes qualitatively (i.e. what proportion of the melt refreezes, is stored, etc, rather than what quantity of water refreezes); the critical values are refering to important points in figure 7, and we now reference this more explicitly in the text. Part of our plan for future work is to decipher the dependence of these transitions on the parameters of the system for specific locations (when numerical values will become more relevant), and to understand the transient response of the system to warming.

20. p 15, l 14: why is storage equal to the flux out? Shouldn't storage be what doesn't leave the system?

    Thanks to the reviewer for noting this ambiguity. Storage in our model refers to water that is stored in the firn or ice, but since the domain of calculation is just the surface region of the ice sheet, such water eventually moves out of the domain if there is continual accumulation (as in the periodic states considered here). Water that passes out of the bottom of the domain is 'stored' in the ice sheet below. (Runoff, on the other hand, comes out of the top of the domain, and is dealt with in the surface boundary conditions). We have amended the text to clarify this.

21. P16 l17 Glacier surface should be Glacier facies, Make sure this is consistent.

    This sentence has been re-worded.

22. p 16, l 27: can they define what surface slope they are referring to that controls "lateral percolation"? Also, I would say lateral flow.

    We meant the slope of the 'water table' in the firn (the top of the fully saturated region which forms above impermeable layers). We have changed the wording to reflect this.